# Sexual Excitation in Young Women with Different Levels of Sexual Satisfaction in Relationships: A Laboratory Study

**DOI:** 10.3390/bs14090769

**Published:** 2024-09-02

**Authors:** María del Mar Sánchez-Fuentes, Ana Álvarez-Muelas, Oscar Cervilla, Reina Granados, Juan Carlos Sierra

**Affiliations:** 1Mind, Brain, and Behavior Research Center (CIMCYC), University of Granada, 18011 Granada, Spain; mmsanchez@ugr.es (M.d.M.S.-F.); alvarezm@ugr.es (A.Á.-M.); ocervilla@ugr.es (O.C.); 2Department of Nursing, Faculty of Health Sciences, University of Granada, 18071 Granada, Spain; reina@ugr.es

**Keywords:** sexual satisfaction, sexual excitation, sexual inhibition, subjective sexual arousal, genital response, young women

## Abstract

Sexual satisfaction is an important dimension of sexual health. Despite there being evidence about its relations with sexual arousal, this association has not been addressed considering arousal as a trait and as a state. Therefore, the goal of this laboratory study was to examine, in young women with different levels of sexual satisfaction in their relationships, sexual arousal as a trait (i.e., propensity for sexual excitation/inhibition) and as a state (i.e., genital response and self-reported sexual arousal to a video with explicit sexual content). The sample was composed of 45 Spanish heterosexual women with a partner, from 18 to 25 years old. In the first phase, we evaluated the propensity for sexual inhibition/excitation and sexual satisfaction of the participants. In the second phase—in the context of a sexuality laboratory—the vaginal pulse amplitude, the rating of sexual arousal, and the rating of genital sensations were evaluated. The participants, distributed in three groups with different levels of sexual satisfaction (*p* < 0.001), were compared based on the different measures of sexual arousal. The results indicated that the group with lower sexual satisfaction, compared to the group with higher sexual satisfaction, reported more sexual inhibition due to the threat of performance failure (*p* = 0.011) and due to the threat of performance consequences (*p* = 0.038). However, no significant differences in sexual arousal status were found between the three groups. In conclusion, the negative association between sexual satisfaction and propensity for sexual inhibition in young women with a partner is supported, but not the positive association between trait/state sexual arousal and sexual satisfaction.

## 1. Introduction

Sexual satisfaction is an essential part of health and is one of the most relevant manifestations of sexual health [1,2]. According to the World Association for Sexual Health (WAS), sexual satisfaction is considered an essential element of both sexual rights and sexual well-being. Furthermore, the WAS expresses the need to strengthen the development of scientific knowledge on this issue [3]. Sexual satisfaction is considered a key factor for quality of life and general well-being [4,5,6,7]. Sexual satisfaction within the framework of couple relationships has been defined as “an affective response arising from one’s subjective evaluation of the positive and negative dimensions associated with one’s sexual relationship” [8] (p. 268). This not only depends on the sexual relationships themselves but also on personal, interpersonal, and sociocultural factors [6]. Relationship satisfaction, of an interpersonal nature, is the variable with the greatest explanatory capacity for sexual satisfaction, so greater relationship satisfaction implies greater sexual satisfaction [9,10,11]. Among the elements inherent to the relationship, its duration stands out due to its negative impact on sexual satisfaction [12,13,14,15,16]. These findings highlight the relevance of the socio–emotional aspects of the couple’s relationship [17]. Although both men and women consider the emotional aspects of their relationships as benefits [18], in the case of women, it seems that they tend to value the emotional aspects associated with relationships as more positive elements in their sexual relationships [10]. A recent study found that interpersonal closeness was one of the main predictors of relationship satisfaction in women, while, in men, the only predictor of relationship satisfaction was sexual satisfaction [19]. 

Sexual satisfaction is associated with the components of the sexual response [20] as a result of adequate sexual functioning [4,21]. In this context, the association of satisfaction with sexual arousal stands out and, with the latter being described as physiological reactions (i.e., genital response) and affective experiences (i.e., assessment of sexual arousal/genital sensations) [22]. 

Sexual arousal can be approached from two complementary perspectives to capture this complexity, recognizing that it can vary both between individuals (as a trait) and within individuals over time or in specific situations (as a state) [23]. Dual Control Model (DCM) explains sexual response, specifically, sexual arousal as a trait, that is, the balance of the dimensions of sexual excitation and inhibition. The excitatory and inhibitory systems are relatively independent and, when they act together, they provide “double control” over the sexual response, as well as over sexual behavior. The DCM proposes that individuals are distributed along a continuum of propensity toward sexual arousal or inhibition [24]. This model, considering these two dimensions as the propensity to become sexually excited/inhibited (trait), has shown its usefulness in understanding sexual dysfunctions. There is scientific evidence that indicates that high levels of inhibition are related to worse sexual functioning [25,26,27,28,29,30,31]. The assessment of these dimensions is performed using instruments such as the Sexual Inhibition/Sexual Excitation Scales-Short Form (SIS/SES-SF) [28,32]. This instrument assesses the propensity for sexual excitation (SES), the propensity for inhibition due to the threat of performance failure (SIS1), and the propensity for sexual inhibition due to the threat of performance consequences (SIS2).

Sexual arousal has also been defined as a state by Janssen (p. 710) who defined it as an “emotional/motivational state that can be activated by internal and external stimuli and that can be inferred from central (including verbal), peripheral (including genital), and behavioral responses (including action tendencies and motor preparation)” [33]. This situational arousal can manifest itself physiologically (e.g., genital response) and subjectively (e.g., self-reported appraisal).

Of the responses elicited, the most specific physiological response to sexual arousal as a state is the genital one [34]. This response can be assessed by self-reports and by using psychophysiological assessment techniques such as photoplethysmography. In this sense, the Information Processing Model of sexual arousal stands out, which differentiates between two stages, the evaluation of the stimulus and the generation stage [35]. Depending on the evaluation of the stimulus, a subjective experience of sexual arousal and sexual response will occur. In this stage, it is evaluated whether the stimulus is pleasurable or not. The generation stage refers to the sexual response [35]. Thus, this model proposes a differentiation of the dimensions of sexual arousal, these being the genital and the subjective.

Physiological sexual arousal in women is a central and peripheral neurophysiological process that results from sexual stimulation and involves bodily responses such as vaginal lubrication, vasocongestion, and vaginal and clitoral congestion [36], a response that is usually assessed by vaginal pulse amplitude [37]. Subjective sexual arousal is defined as the perception of sexual arousal at a psychological level [22], so, self-reports are used for its assessment [22,38]. In the case of women, it is essential to consider both dimensions, since the study of the relationship between these two dimensions of sexual arousal (subjective and physiological) has led to a wide variety of results. Some research concludes that there is low or no concordance in women [37,39,40], while others conclude the opposite [34,41].

Considering sexual arousal as a trait, the scarce evidence that has related DCM dimensions to sexual satisfaction in the context of couple relationships is inconclusive. Specifically, while some studies have observed no relationship between propensity for sexual excitation and sexual satisfaction in women [28,42], others have pointed to a negative relationship between both variables [43]. The discrepancies between these studies could be explained, in part, by the way in which sexual satisfaction is assessed, whether it has been considered as a dimension of sexual functioning [28] or as a more global standardized measure of satisfaction [42,43]. Regarding the relationship between sexual inhibition and sexual satisfaction, the results are more consistent, with the finding that the propensity for sexual inhibition due to threat of performance failure [28,42,43], and inhibition due to threat of performance consequences [28], have been found to be associated with lower sexual satisfaction. Considering sexual arousal as a state, evidence is also limited. In the case of women, feeling sexually aroused by one’s romantic partner was positively related to one’s own sexual satisfaction, whereas feeling sexually aroused by people other than one’s romantic partner was negatively associated with one’s own sexual satisfaction [44]. Furthermore, similar levels of arousal between both members of a couple have been related to result in more sexual satisfaction in women than when there are discrepancies [45,46]. However, it should be noted that in these studies sexual arousal refers to arousal during sexual activity with a partner, an aspect that highlights the relevance of deepening the relationship between arousal and sexual satisfaction from a broader perspective, considering standardized measures. In addition, there was no evidence of the relationship with physiological measures of sexual arousal, such as genital response. 

In conclusion, sexual arousal could be considered as a trait (i.e., propensity for sexual excitation/inhibition, as proposed by the DCM) and a state (i.e., genital response and subjective evaluation of a sexual stimulus). To our knowledge, this double approach to sexual arousal has not been considered when examining its relationship with sexual satisfaction.

Therefore, the goal of the present study was to examine, in young women with different levels of sexual satisfaction in their relationships, sexual arousal as a trait (i.e., propensity for sexual excitation/inhibition) and as a state (i.e., sexual arousal when faced with a video with explicit sexual content, evaluated objectively through photoplethysmography and subjectively through self-reports). We expect that women who are more sexually satisfied in relationships compared to less satisfied women would (1) report lower propensity for sexual excitation [43] or not differ from each other [28,42], (2) report lower propensity for sexual inhibition [28,42,43], and (3) experience lower sexual arousal in the laboratory context (i.e., lower genital responsiveness and subjective sexual arousal to sexual stimuli) due to the presentation of sexual content showing someone other than her own partner [44].

## 2. Materials and Methods

The sample was composed of 45 Spanish heterosexual women with a partner. The age range was 18–25 years (*M* = 20.67; *SD* = 1.93). The exclusion criteria were as follows: (a) having medical problems, sexual dysfunction, and/or psychological disorders; (b) taking medication that could interfere with sexual response; (c) drugs and/or alcohol abuse; (d) a history of sexual abuse. The mean of the duration of relationships was 26.96 months (*SD* = 19.35), the age of the first sexual relationship was 16.22 (*SD* = 1.35), and number of sexual partners was 4.58 (*SD* = 4.98).

### 2.1. Instruments and Materials

We used the Socio-demographic and Sexual History Questionnaire. It was designed to assess sex, age, nationality, sexual orientation, relationship duration, medical/psychological/sexual problems, medical treatments, drug/alcohol use, and sexual abuse history.

We used the Spanish version of the Global Measure of Relationship Satisfaction (GMREL) [47,48]. It evaluates satisfaction with a partner by five items answered on seven-point bipolar subscales (good–bad, pleasant–unpleasant, positive–negative, satisfying–unsatisfying, valuable–worthless). Its Cronbach’s alpha coefficients are 0.94 for women [48]. In the sample of this study, the McDonald’s omega was 0.95.

We used the Spanish version of the Global Measure of Sexual Satisfaction (GMSEX) [47,48]. It evaluates overall sexual satisfaction in a relationship context by five items answered on seven-point bipolar subscales (good–bad, pleasant–unpleasant, positive–negative, satisfying–unsatisfying, valuable–worthless). Its Cronbach’s alpha coefficients are 0.93 for women [48]. In the sample of this study, the McDonald’s omega was 0.94.

We used the Spanish version of the Sexual Inhibition/Sexual Excitation Scales-Short Form (SIS/SES-SF) [28,32]. It assesses the propensity for sexually excited/inhibited by 14 items answered on a four-point Likert-type scale from 1 (strongly agree) to 4 (strongly disagree). These items are distributed on three subscales: sexual excitation (SES); inhibition due to the threat of performance failure (SIS1); inhibition due to the threat of performance consequences (SIS2). The original scale has adequate reliability, with Cronbach’s alpha coefficients ranging between 0.60 and 0.72. Their scores show good evidence of validity [31]. In the sample of this study, the McDonald’s omega was 0.84 for SES, 0.69 for SIS1, and 0.82 for SIS2. 

We used the Spanish version of the Rating of Sexual Arousal (RSA) [22,38]. Its five items evaluate subjective sexual arousal: (1) overall level of sexual arousal; (2) intensity of genital sensations; (3) sensation of warmth experienced; (4) non-genital physical sensations; (5) level of sexual concentration, answered on a seven-point Likert-type scale from 1 (no sexual arousal at all) to 7 (extremely sexually aroused). The original scale obtained a Cronbach’s alpha score of 0.90 [22]. Its measures show adequate validity evidence [49]. In the sample of this study, the McDonald’s omega was 0.84.

We used the Spanish version of the Rating of Genital Sensations (RGS) [22,38]. It assesses the level of genital sensation through a checklist scale from 1 (no genital sensations) to 11 (multiple orgasms). It has shown adequate evidence of validity [49].

The Biopac MP 150 polygraph with 16 channels (Biopac Systems Inc., Goleta, CA, USA) is used by the AcqKnowledge software 5.0 for psychophysiological data acquisition and processing. Vaginal photoplethysmography (Biopac amplifier PPG100C and vaginal transducers) was used. 

Visual stimuli. A 3-minute neutral content film (nature documentary) and a 3-minute sexual film (sexually explicit heterosexual video in which a couple has a sexual relationship including oral sex and vaginal intercourse). A pilot study was carried out to ensure that the sexual film elicited sexual arousal.

### 2.2. Procedure

Women were invited to participate voluntarily and without compensation in this study through the dissemination methods of the University of Granada. Interested volunteers accessed an online survey to review inclusion/exclusion criteria and accepted informed consent with the objective of this study. Eligible participants were contacted and invited to the experimental task in the human sexuality laboratory. The appointment was not during menstruation, under the instruction to abstain from caffeine, alcohol, and dyadic and solitary sexual activity 24 h before the experimental session. In the laboratory, participants signed an informed consent form, with guarantees of anonymity and confidentiality. Before the experimental task, in the first phase, participants reported their propensity for sexual inhibition/excitation and sexual satisfaction in their relationships in an online survey while they were alone in the room. A woman researcher explained the experimental task and devices to record the genital response before she left the room. In all cases, the experiment was carried out in the same conditions of temperature, light, and humidity. Before the experiment, the participants had five minutes to rest and to adapt to the experimental conditions and devices. The task consisted of presenting the neutral video and the sexual video featuring someone other than her own partner, with a duration of three minutes each. During the viewing, the genital response was recorded with vaginal photoplethysmography (Biopac amplifier PPG100C and vaginal transducers). The genital response (i.e., vaginal pulse amplitude) from the difference between the scores of the explicit sexual stimulus and the baseline stimulus were calculated. At the end of the videos, the participants answered the RSA and RGS scales. This study was previously approved by the Ethics Committee on Human Research of the University of Granada.

### 2.3. Data Analysis

The R^®^ program (version 3.6.3) [50] with the RStudio^®^ interface (version 1.2.5042) [51] was used. The missForest package was employed for missing data (version 1.4) [52], and the Psych package (version 1.9.12.31) [53] was used to calculate the McDonald’s omega value. The other statistical analyses were performed with IBM^®^ SPSS^®^ v.22. Firstly, descriptive statistics and bivariate correlations of the study variables were examined (i.e., sexual satisfaction, relationship satisfaction, propensity for sexual excitation/inhibition, genital response, RSA, and RSG). According to the scores of the Spanish Global Measure of Sexual Satisfaction (GMSEX) [48], the sample was divided into terciles. Group 1 refers to the tercile with low scores, group 2 refers to the tercile with average scores in sexual satisfaction, and group 3 refers to the tercile with high scores in sexual satisfaction. By univariate analyses, the scores of sexual satisfaction were compared in the three groups, with relationship satisfaction and relationship duration introduced as covariables (ANCOVA). Following this, a *t*-test for related samples was calculated for examining the differences in sexual arousal as a state (i.e., genital response, RSA, and RSG) between the scores of the explicit sexual stimulus (i.e., viewing sexually explicit heterosexual video in which a couple has sexual relationship including oral sex and vaginal intercourse) and the baseline stimulus (i.e., viewing nature documentary). Finally, to examine the relation of sexual arousal measures with sexual satisfaction in their relationships, multivariate analyses (MANOVA) comparing the three terciles of sexual satisfaction (as independent variable) in propensity for sexual excitation/inhibition, genital response, RSA and RGS (as dependent variables) were conducted.

## 3. Results

Descriptive statistics of sexual satisfaction, relationship satisfaction, propensity for sexual excitation/inhibition, genital response, rating of sexual arousal, and genital sensations are shown in Table 1. The bivariate correlations between all study variables are shown in Table 2.

The mean was 25.08 (*SD* = 4.29) for low scores in the sexual satisfaction group, 31.67 (*SD* = 1.24) for average scores in the sexual satisfaction group, and 34.8 (*SD* = 0.41) for high scores in the sexual satisfaction group. In the univariate contrast test (ANCOVA), results showed significant differences in sexual satisfaction in their relationships between the three sexual satisfaction in their relationships groups (*F* = 42.37; *p* < 0.001; η_p_^2^ = 0.68), while controlling for relationship satisfaction (*F* = 18.30; *p* < 0.001). There was no significant effect of the relationship duration as a covariable (*F* = 0.069; *p* = 0.795).

For sexual arousal as a state, during the baseline stimulus (viewing a nature documentary), the mean was 0.04 (*SD* = 0.03) for genital response, 7.02 (*SD* = 2.57) for RSA, and 1.09 (*SD* = 0.36) for RGS. During the explicit sexual stimulus (viewing a sexually explicit heterosexual video in which a couple has a sexual relationship including oral sex and vaginal intercourse), the mean was 0.10 (*SD* = 0.07) for genital response, 19.84 (*SD* = 4.40) for RSA, and 3.77 (*SD* = 1.41). There were differences in these measures—genital response (*t* = 8.104; *p* < 0.001), RSA (*t* = 17.790; *p* < 0.001), and RGS (*t* = 13.079; *p* < 0.001)—between measures of explicit sexual stimulus and the baseline stimulus. Then, in the multivariate contrast test (MANOVA) for examining the differences in sexual arousal measures as a trait (i.e., propensity for sexual excitation/inhibition) and as a state (i.e., genital response, rating of sexual arousal, and rating of genital sensations), results showed an effect of the sexual satisfaction group (Roy’s Largest Root = 0.45; *F* = 2.90; *p* = 0.020). Significant differences were observed between sexual satisfaction groups in inhibition due to the threat of performance failure (*F* = 4.79; *p* = 0.013), and inhibition due to the threat of performance consequences (*F* = 3.68; *p* = 0.034). The group with low scores of sexual satisfaction in their relationship reported higher scores of inhibition due to the threat of performance failure (*p* = 0.011) and inhibition due to the threat of performance consequences (*p* = 0.038) than the group with high scores of sexual satisfaction in their relationship. Table 3 shows the differences in sexual arousal measures.

## 4. Discussion

The present study aimed to provide evidence about the association of sexual satisfaction in their relationships with sexual arousal as a trait (i.e., propensity for sexual excitation/inhibition) and as a state (i.e., genital response and self-reported sexual arousal when watching a sexual video) in young women who maintain a heterosexual relationship. As suggested by the results obtained, there are only differences in the propensity for sexual inhibition between women with lower and higher sexual satisfaction in their relationships, with the latter presenting lower inhibition.

Firstly, the results showed a positive association between sexual satisfaction and relationship satisfaction, congruent with previous evidence [9,10,11]. Meanwhile, relationship satisfaction was negatively associated with genital response, which seems to support the findings of Lawless et al. [44] that feeling sexually aroused by people other than one’s own romantic partner, such as viewing an explicit sexual stimulus, could be associated negatively with one’s own sexual satisfaction. The negative correlation between SES and SIS2, and positive between SIS1 and SIS2, seem to demonstrate the relative independence between both sexual excitation/inhibition systems [23]. A positive association between self-reported sexual arousal (i.e., RSA and RSG) is noted, although these variables are not related to the genital response, which could reflect the absence of sexual concordance (i.e., the association between the genital and subjective response) in women [37,41,49,54,55]. The lack of association between trait (i.e., SES, SIS1, and SIS2) and state (genital response, RSA, and RSG) also seems to support the two complementary perspectives approaches to examining the complexity of sexual arousal [23].

Secondly, the results indicate significant differences only in the two dimensions of sexual inhibition: inhibition due to the threat of performance failure and inhibition due to the threat of performance consequences. Specifically, the group with low scores in sexual satisfaction in their relationship obtained higher scores in inhibition due to the threat of performance failure and inhibition due to the threat of performance consequences in comparison with the group with high scores of sexual satisfaction in their relationship. This result, congruent with the observed correlations, seems to indicate that sexual satisfaction in their relationships is more related to trait sexual arousal (in this case, the propensity for sexual inhibition) than to state sexual arousal (i.e., genital response and the appraisal of sexual arousal experienced in a certain moment). It seems logical to think that sexual satisfaction in their relationship, being a concept that encompasses a broad reality and not so much a specific situation (i.e., a global description of a sexual relationship with the partner), is related to a greater extent with a trait, or form of being of the person, than with what that person may experience in a specific situation, such as exposure to a sexual film. In the case of women, it has been described that the experience of sexual arousal would be more determined by a general disposition than by a particular response, such as the genital response [56]. It has been pointed out that sexuality in women would be conditioned by a diversity of variables [57], a fact that could partly explain the greater relevance of more global and dispositional psychosexual dimensions, rather than specific experiences.

The differences observed in sexual inhibition and its negative correlation with sexual satisfaction, as well as the absence of differences and relationship with sexual excitation, reflect the importance of the propensity for sexual inhibition for women [28,32,58,59]. Previous evidence indicates that more general dimensions, such as a propensity for sexual excitation, as opposed to more contextual or situational dimensions, such as inhibition due to the threat of performance failure or consequences, may be of lesser importance in women’s sexuality [28]. Dèttore et al. [60], when comparing women with anxiety disorders and sexual problems with healthy women, found that they differed only in sexual inhibition due to the threat of performance failure and sexual inhibition due to the threat of performance consequences, and not in sexual excitation. In general, the scientific literature indicates that the excitatory trait is usually associated to a lesser extent than the inhibitory trait with sexual problems in women [29]. Previous studies have already highlighted the negative relations between sexual inhibition due to the threat of performance failure [28,42,43] and sexual inhibition due to the threat of performance consequences [28] with female sexual satisfaction. Sexual inhibition due to the threat of performance failure is considered a negative predictor of sexual satisfaction and different dimensions of sexual functioning in women [28,29,31,43,61,62,63,64]. In the heterosexual relationship context, vaginal intercourse is the most highly valued sexual activity [65], thus, women might experience more pressure to experience orgasm through penetration alone [66,67]. In this line, concerns related to the sexual act could also reinforce the processes of sexual inhibition due to the threat of performance failure and may affect their levels of sexual satisfaction in the relationship. On the other hand, it was expected that the propensity to be sexually inhibited in situations where pain, exposure, or the risk of unwanted pregnancy is anticipated would be associated with more sexual dissatisfaction.

This study is not free of limitations that must be considered for the generalization of the results. First, laboratory studies often have reduced ecological validity because the experimental conditions can differ significantly from the real-world context in which arousal is experienced. Secondly, future studies should expand the sample and consider greater diversity in sociodemographic characteristics (e.g., age or sexual orientation) as well as relationship characteristics (e.g., relationship duration, cohabitation, exclusivity, or the frequency of sexual activity). Although the groups were significantly different in sexual satisfaction in the relationship, it is suggested to have clinical samples that report sexual dissatisfaction. Likewise, future research could consider variables specific to the couple’s relationship which could affect sexual satisfaction [57]. For example, it has been noted that women tend to evaluate their sexual satisfaction considering that of their partners [10,57,68,69]. A variable to also consider in future studies, given that it negatively affects sexual satisfaction, is the experienced intimate partner violence [70,71,72,73], including with the current partner [74]. 

## 5. Conclusions

To conclude, this research provides evidence of the association between satisfaction and sexual arousal in women. Specifically, it points out the negative association between sexual satisfaction in the relationship and the propensity for sexual inhibition due to the threat of performance failure and the inhibition due to the threat of performance consequences. This finding could have implications, both for future research and for clinical practice, by highlighting the consideration of sexual arousal as a trait and state, the relative independence of sexual excitation/inhibition systems, and the negative role of the propensity for sexual inhibition in women’s sexual satisfaction in the context of a relationship. Among the clinical strategies that are derived, the importance of personal variables (i.e., propensity for sexual inhibition) for women’s sexual satisfaction is emphasized. As in previous investigations, multicomponent models are considered essential for the study of sexual satisfaction, such as the Ecological Theory of Human Development of Bronfenbrenner [1,75,76,77,78]. Also, it is intended to pay attention to the processes of sexual inhibition in women, considering the possible role of gender roles [79]. In this regard, propensity for sexual inhibition could be considered an adaptative mechanism with the purpose of hindering the sexual response to stimuli in situations evaluated as threatening [80], and so it may be necessary to evaluate the role of women´s sexual inhibition in the context of sexual relationships and to provide strategies in sex therapy aimed at promoting satisfactory sexuality in the context of a couple’s relationship.

## Figures and Tables

**Table 1 behavsci-14-00769-t001:** Descriptive statistics of the study variables.

	*M*	*SD*	Range Study	Range Scales
Sexual satisfaction	30.96	4.46	15–35	5–35
Relationship satisfaction	31.16	4.42	17–35	5–35
SES	15.60	2.82	11–23	6–24
SIS1	8.20	1.84	4–12	4–16
SIS2	12.24	2.31	6–16	4–16
Genital response	0.06	0.05	0–0.21	-
RSA	19.84	4.40	12–28	5–35
RGS	3.80	1.41	1–7	1–11

Note. SES: Sexual excitation; SIS1: Inhibition due to the threat of performance failure; SIS2: Inhibition due to the threat of performance consequences; RSA: Rating of Sexual Arousal; RGS: Rating of Genital Sensations. A range of scales is included.

**Table 2 behavsci-14-00769-t002:** Bivariate correlations between the study variables.

	Relationship Satisfaction	SES	SIS1	SIS2	Genital Response	RSA	RSG
Sexual satisfaction	0.65 **	0.19	−0.35 *	−0.31 *	−0.18	0.03	0.15
Relationship satisfaction		0.07	−0.13	−0.12	−0.32 *	−0.05	−0.01
SES			−0.20	−0.43 **	0.02	0.24	0.13
SIS1				0.54 **	0.09	0.08	−0.11
SIS2					−0.03	0.10	0.16
Genital response						0.03	−0.08
RSA							0.61 **

Note. SES: Sexual excitation; SIS1: Inhibition due to the threat of performance failure; SIS2: Inhibition due to the threat of performance consequences; RSA: Rating of Sexual Arousal; RGS: Rating of Genital Sensations; * *p* < 0.05; ** *p* < 0.01.

**Table 3 behavsci-14-00769-t003:** Differences between groups in sexual satisfaction.

	Low Scores in Sexual Satisfaction Group	Average Scores in Sexual Satisfaction Group	High Scores in Sexual Satisfaction Group			
	*n* = 12	*n* = 18	*n* = 15			
	*M (SD)*	*M (SD)*	*M (SD)*	*F*	*p*	η_p_^2^
SES	14.92 (2.47)	15.72 (3.30)	16 (2.54)	0.50	0.606	
SIS1	9.41 (1.38) _a_	8.06 (2.04)	7.4 (1.45) _a_	4.79	0.013	0.19
SIS2	13.67 (1.61) _a_	11.94 (2.62)	11.47 (1.96) _a_	3.68	0.034	0.15
Genital response	0.07 (0.03)	0.06 (0.06)	0.05 (0.05)	0.71	0.496	
RSA	21.33 (4.36)	19.11 (4.34)	19.53 (4.52)	0.97	0.386	
RGS	3.58 (1.16)	4 (1.57)	3.73 (1.44)	0.33	0.721	

Note. SES: Sexual excitation; SIS1: Inhibition due to the threat of performance failure; SIS2: Inhibition due to the threat of performance consequences; RSA: Rating of Sexual Arousal; RGS: Rating of Genital Sensations; η_p_^2^: partial eta squared. Subscript letters denote the mean difference between groups at the 0.05 level; adjustment for multiple comparisons by Bonferroni correction.

## Data Availability

The data presented in this study are available on request from the corresponding author. The data are not publicly available due to privacy.

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
