# Peer review of "Sexual Excitation in Young Women with Different Levels of Sexual Satisfaction in Relationships: A Laboratory Study"

_behavsci, 2024, doi:10.3390/bs14090769_

Round 1

Reviewer 1 Report

Comments and Suggestions for Authors

Thank you for the opportunity to review this interesting manuscript. It reports on a relevant study among young women on associations of trait and state dimensions of sexual arousal with their sexual satisfaction. While reading the paper, a number of critical points came up, that deserve attention. Below I will describe these points in the order in which they appear in the manuscript.

Abstract

Page 1, line 17-18: “In the first phase, we evaluated the propensity 17 for sexual inhibition/excitation of the participants” Was sexual satisfaction also measured in this phase? This should be added here.

Introduction

Here the authors reverse the WAS statement to which they refer. The text referred to is the WAS declaration on sexual pleasure, in which the following statement can be found: "Sexual pleasure is the physical and/or psychological satisfaction and enjoyment derived from shared or solitary erotic experiences …”. The reverse claim is not necessarily following from this statement. In other literature it is made clear that sexual satisfaction is determined by many more factors than sexual pleasure alone. This position is also taken by the authors themselves from line 37 onward.

The introduction pays little attention to introducing the (state) physiological and subjective aspects of sexual arousal and their interrelationships. Are both aspects equally important for sexual satisfaction? To what extent do these aspects overlap or affect each other's influence on sexual satisfaction? Why is it not sufficient to use either aspect, and which of the two is most relevant in the context of this study? This introduction is relevant to the choice of using both aspects in operationalizing sexual arousal.

The research question is not translated into testable hypotheses, although the introduction provides ample support, both theoretically and empirically, to do so. However, in the Results paragraph, the authors report according to a hypothesis-testing design. Not testing specific and focused hypotheses means that the study is exploratory in nature, which should be explicitly stated in the introduction and abstract to correctly guide reader expectations.

Method

What is the rationale for splitting up the sample in three subgroups on sexual satisfaction, given that this operation reduces data variability and statistical power for further analyses? Although the effects were – apparently - large enough to be identified even after categorization, this would be preferable.

Results

Descriptive statistics of the key study variables and their bivariate correlations are lacking. In addition to providing the reader with the opportunity to gauge the raw characteristics of these variables and their bivariate associations, this is also necessary to underpin the decisions to include covariables in hypothesis testing.

This lack of descriptive statistics makes it impossible to evaluate the data on state sexual arousal. How large was the variability of these variables at baseline and during the experimental conditions? Were there effects of erotic stimulation? To calculate genital and subjective sexual arousal scores for hypothesis testing, were difference scores between baseline and experimental condition preferable over analysis of covariance (See van Breukelen, 2013)?

Page 4, lines 169-170: … WHILE controlling FOR relationship satisfaction …

Page 4, line 177: Please make explicit which dependent and independent variables were included in multivariate testing as required in APA7. Were trait and state measures of sexual arousal all included in the same MANOVA?

Why was relationship satisfaction not included in multivariate testing, given the large proportion of explained variance of this variable for sexual satisfaction, at least for women, and its unknown interrelationships with sexual excitation/inhibition proneness? The exploratory research question makes a strong case for this.

van Breukelen, G. J. P. (2013). ANCOVA versus CHANGE from baseline in nonrandomized studies: The difference. Multivariate Behavioral Research, 48(6), 895–922. Doi: 10.1080/00273171.2013.831743

Comments on the Quality of English Language

N/A

Author Response

Thank you for the opportunity to review this interesting manuscript. It reports on a relevant study among young women on associations of trait and state dimensions of sexual arousal with their sexual satisfaction. While reading the paper, a number of critical points came up, that deserve attention. Below I will describe these points in the order in which they appear in the manuscript.

>>Response: We thank you for your evaluation of our work and, especially, for all your indications to improve the manuscript. Below we respond to each one of them.

Abstract

Page 1, line 17-18: “In the first phase, we evaluated the propensity 17 for sexual inhibition/excitation of the participants” Was sexual satisfaction also measured in this phase? This should be added here.

>>Response: Thank you for your recommendation. We have included the sexual satisfaction measure in the first phase.

Introduction

Here the authors reverse the WAS statement to which they refer. The text referred to is the WAS declaration on sexual pleasure, in which the following statement can be found: "Sexual pleasure is the physical and/or psychological satisfaction and enjoyment derived from shared or solitary erotic experiences …”. The reverse claim is not necessarily following from this statement. In other literature it is made clear that sexual satisfaction is determined by many more factors than sexual pleasure alone. This position is also taken by the authors themselves from line 37 onward.

>>Response: Thank you for your indication. We fully agree with your correction and have therefore removed this statement.

The introduction pays little attention to introducing the (state) physiological and subjective aspects of sexual arousal and their interrelationships. Are both aspects equally important for sexual satisfaction? To what extent do these aspects overlap or affect each other's influence on sexual satisfaction? Why is it not sufficient to use either aspect, and which of the two is most relevant in the context of this study? This introduction is relevant to the choice of using both aspects in operationalizing sexual arousal.

>>Response: Thank you very much for your indication. We have included in the Introduction section more information on sexual arousal, differentiating between trait sexual arousal (DCM) and state sexual arousal (in the laboratory context). We also report on the lack of concordance that seems to exist in the case of women between the subjective (self-reported) and objective (genital response) dimensions of sexual arousal.

The research question is not translated into testable hypotheses, although the introduction provides ample support, both theoretically and empirically, to do so. However, in the Results paragraph, the authors report according to a hypothesis-testing design. Not testing specific and focused hypotheses means that the study is exploratory in nature, which should be explicitly stated in the introduction and abstract to correctly guide reader expectations.

>>Response: Thank you very much for your indication. We have included hypothesis to increase the coherence of the design.

Method

What is the rationale for splitting up the sample in three subgroups on sexual satisfaction, given that this operation reduces data variability and statistical power for further analyses? Although the effects were – apparently - large enough to be identified even after categorization, this would be preferable.

>>Response: We find your comment about the analyses very interesting. As you know, one limitation of laboratory studies is the difficulty in obtaining many participants. This fact makes it difficult or impossible to perform some analyses. Previous studies with similar samples and methodology have opted to divide the sample into groups with high and low scores on certain variables and then compare them on the variables under study (e.g., Bradford & Meston, 2006; Janssen et al., 2002), which is the reason that led us to perform this type of analysis. Using G*Power, we calculated the total sample for this type of analysis and found that we have a sufficient sample to perform a MANOVA of three groups and six measurements.

Bradford, A., & Meston, C. M. (2006). The impact of anxiety on sexual arousal in women. Behaviour Research and Therapy, 4488), 1067-1077. https://doi.org/10.1016/j.brat.2005.08.006

Janssen, E., Vorst, H., Finn, P., & Bancroft, J. (2002). The sexual inhibition (SIS) and sexual excitation (SES) scales: II. Predicting psychophysiological response patterns. The Journal of Sex Research, 39(2), 127-132. https://doi.org/10.1080/00224490209552130

 Results

Descriptive statistics of the key study variables and their bivariate correlations are lacking. In addition to providing the reader with the opportunity to gauge the raw characteristics of these variables and their bivariate associations, this is also necessary to underpin the decisions to include covariables in hypothesis testing.

>>Response: Thank you for your recommendation. We have now included two new tables: (1) descriptives of the study variables and (2) bivariate correlations between all variables.

This lack of descriptive statistics makes it impossible to evaluate the data on state sexual arousal. How large was the variability of these variables at baseline and during the experimental conditions? Were there effects of erotic stimulation? To calculate genital and subjective sexual arousal scores for hypothesis testing, were difference scores between baseline and experimental condition preferable over analysis of covariance (See van Breukelen, 2013)?

van Breukelen, G. J. P. (2013). ANCOVA versus CHANGE from baseline in nonrandomized studies: The difference. Multivariate Behavioral Research, 48(6), 895–922. https://doi.10.1080/00273171.2013.831743

>>Response: We thank you again for your comment and include this information. We calculated the genital response, RSA and RGS from the difference between the scores of the explicit sexual stimulus (viewing sexually explicit heterosexual video in which a couple has sexual relationship including oral sex and vaginal intercourse) and the baseline stimulus (viewing nature documentary). This information is now included in the Data Analysis subsection and before the MANOVA (Results subsection).

Page 4, lines 169-170: … WHILE controlling FOR relationship satisfaction …

>>Response: Thank you, done.

Page 4, line 177: Please make explicit which dependent and independent variables were included in multivariate testing as required in APA7. Were trait and state measures of sexual arousal all included in the same MANOVA?

>>Response: Thank you for your comment, it is useful for clarifying the analysis. We included trait and state measures of sexual arousal. This information is now provided in the manuscript at the end of the Data Analysis subsection.

 Why was relationship satisfaction not included in multivariate testing, given the large proportion of explained variance of this variable for sexual satisfaction, at least for women, and its unknown interrelationships with sexual excitation/inhibition proneness? The exploratory research question makes a strong case for this.

>>Response: Considering that the aim of the study is to compare the different dimensions of sexual arousal among three groups of women with different levels of sexual satisfaction, and that in the configuration of these three groups the effect of satisfaction with the partner relationship was controlled for, we did not consider it necessary to include this last variable in the multivariate analyses. Moreover, in the univariate analyses for examining the differences in the scores of sexual satisfaction were compared into the three groups, there was no significant effect of the relationship duration as a covariable (F = 0.069; p = .795).

Again, we would like to sincerely thank the reviewer for the time spent in reviewing our manuscript and the constructive criticism to improve our work. Thank you very much!

Reviewer 2 Report

Comments and Suggestions for Authors

I enjoyed reading this paper and think the topic is significant. But I recommend several changes to be published.

Title and Abstract

-  Reading the title and abstract of this paper, it needs to be clarified that this study has a laboratory and experimental design. I recommend that the authors make this more explicit.

- Also, the abstracts need a conclusion from the main results.

Introduction

-  The introduction needs improvement on several points, and I recommend the authors change it to contextualise this study and its objectives more specifically. The authors didn’t demonstrate through the literature presentation the importance of this study or what this study brings to the state of the art. This has a significant impact on the way the authors explain the objective. I also wonder if it isn’t possible to think of more research questions if hypotheses aren’t likely to be present. Also: 

- The authors explain sexual satisfaction in detail, highlighting relationship satisfaction and couple factors. If one of the main variables is sexual satisfaction with relationships, this idea must be evident in the paper title.

- The authors then create a paragraph that tries to relate sexual satisfaction with sexual arousal. This paragraph needs to be completed, and the literature should be explored more. On page 2, lines 55-57, I think the association between satisfaction and sexual arousal is unclear. Also, the link between the following two studies needs to be clarified. So, I recommend the authors provide a more detailed description of the studies and the processes that may allow this interconnection.

- They then explain the DCM and briefly refer to some results regarding sexual satisfaction. I think that the authors should be more explicit with these studies’ results and more precise about the sexual satisfaction variable. Is it always measured in the context of a relationship? Do the studies have an experimental design?

          -     Finally, the authors present sexual arousal as a trait and as a state. What supports the authors that this is the best way to measure trait and state sexual arousal? My primary concern is with the state being measured by the SIS/SES because I know that the instructions to answer the items don’t specify a window of time, and participants can respond considering their more recent experience and not in general and through time. I recommend that the authors explain their rationale and indicate support from the literature to explain these main decisions.

Materials and methods

-   Regarding the sample, did the authors consider a minimum time in the relationship as inclusion criteria?

-  It would also be interesting to know what kind of relationship these women have and the frequency of sexual activity.

-  I recommend that the authors include biopac and visual stimuli in the procedures.

-  I wonder if the visual stimuli, namely the sexual one, were previously validated.

-    Did the team consider allowing more time for the participant to rest before the experiment, or thought of having more time in the documentary film? After all the procedures of being in the lab and with the researcher, more than 3 minutes of a neutral film seems necessary. This may have a strong influence on the excitation response. I would like to know if this is considered one of the study’s limitations.

-  The procedures must be clarified when the participants fill out the SIS/SES, GMREL, and GMSEX. Although this may be at the beginning of the online survey, it must be clarified.

-   I recommend the authors explain how the genital response was calculated.

Discussion

-  The discussion becomes more difficult after line 227. The authors are sometimes redundant.

-  The authors should be more coherent in the use of the terms sexual satisfaction and sexual satisfaction in their relationships throughout the paper.

-  Also, it is not clear why excitatory traits are not so related to sexual satisfaction. Clinical practice gives us other information if the literature fails to show this relationship. Also, the link of excitation with orgasm doesn’t explain the results and possible processes that underlie these results. I suggest that the authors go further in explaining these results, considering other processes related to inhibitory and excitation responses.

-   I recommend adding other limitations, namely the ones related to the use of an experimental setting on the sexual response, the consideration of the trait, and the state decisions of the research team.

-   The conclusion doesn’t explain why this study is essential, as the introduction does. I recommend the authors be more specific about how this study may contribute to further studies and, more specifically, to clinical interventions. Based on these results, what clinical strategies does the team suggest?

Author Response

I enjoyed reading this paper and think the topic is significant. But I recommend several changes to be published.

>>Response: We thank you for your revision of our work and, especially, for all your indications to improve the manuscript. Below we respond to each one of them.

Title and Abstract

-  Reading the title and abstract of this paper, it needs to be clarified that this study has a laboratory and experimental design. I recommend that the authors make this more explicit.

>>Response: Thank you very much for your suggestion.  The term "laboratory study" has been included in the title and abstract.

- Also, the abstracts need a conclusion from the main results.

>>Response: Thank you for the suggestion, the abstract has been modified.

Introduction 

-  The introduction needs improvement on several points, and I recommend the authors change it to contextualise this study and its objectives more specifically. The authors didn’t demonstrate through the literature presentation the importance of this study or what this study brings to the state of the art. This has a significant impact on the way the authors explain the objective. I also wonder if it isn’t possible to think of more research questions if hypotheses aren’t likely to be present.

>>Response: We appreciate your recommendation to better contextualize the study. The Introduction has been modified and we hope that these changes will meet your expectations.

- The authors explain sexual satisfaction in detail, highlighting relationship satisfaction and couple factors. If one of the main variables is sexual satisfaction with relationships, this idea must be evident in the paper title.

>>Response: Thank you for your indication. We have included “sexual satisfaction in relationships” in the title to make it more evident.

 - The authors then create a paragraph that tries to relate sexual satisfaction with sexual arousal. This paragraph needs to be completed, and the literature should be explored more. On page 2, lines 55-57, I think the association between satisfaction and sexual arousal is unclear. Also, the link between the following two studies needs to be clarified. So, I recommend the authors provide a more detailed description of the studies and the processes that may allow this interconnection.

>>Response: Thank you very much for your recommendation. We have improved the wording regarding the explanation of sexual satisfaction and arousal. We also explain the results of the other two studies mentioned.

- They then explain the DCM and briefly refer to some results regarding sexual satisfaction. I think that the authors should be more explicit with these studies’ results and more precise about the sexual satisfaction variable. Is it always measured in the context of a relationship? Do the studies have an experimental design?

>>Response: Thank you very much for the indication. We fully agree with the reviewer, therefore, we have expanded the presentation of the papers that have related the dimensions of the DCM model to sexual satisfaction.

 - Finally, the authors present sexual arousal as a trait and as a state. What supports the authors that this is the best way to measure trait and state sexual arousal? My primary concern is with the state being measured by the SIS/SES because I know that the instructions to answer the items don’t specify a window of time, and participants can respond considering their more recent experience and not in general and through time. I recommend that the authors explain their rationale and indicate support from the literature to explain these main decisions.

>>Response: Thank you for your insightful feedback. We advocate the measurement of sexual arousal as both a trait and a state to capture the complexity of sexual arousal, recognizing that it can vary both between individuals (as a trait) and within individuals over time or in specific situations (as a state).

Regarding your concern about the assessment of trait arousal, we inform you that the instrument used to assess the propensity for sexual arousal/inhibition (trait arousal) is the SIS/SES-SF, scales developed and validated in different cultures for that purpose (see Janssen & Bancroft, 2023). The instructions for the scales indicate that responses to the items should be contextualized in general (not in relation to the last situation).

Janssen, E., & Bancroft, J. (2023). The Dual Control Model of Sexual Response: A scoping review, 2009–2022. The Journal of Sex Research, 60(7), 948-968. https://doi.org/10.1080/00224499.2023.2219247

 Materials and methods

-   Regarding the sample, did the authors consider a minimum time in the relationship as inclusion criteria?

>>Response: We did not consider a minimum relationship duration as an inclusion criterion. The range in months of relationship duration was 1-66 (M = 26.96, SD = 19.35). Since this variable could influence sexual satisfaction and satisfaction with the relationship, we have included this issue as a limitation of the study.

 -  It would also be interesting to know what kind of relationship these women have and the frequency of sexual activity.

>>Response: Thank you for your interesting appreciation. In this study, we ensured that all women had previous sexual experience. Unfortunately, we do not have information about the type of relationship (e.g., cohabitation, exclusivity, etc.) or the frequency of sexual activity of these women. The variables that you mentioned are interesting, and we will consider them for future studies. We have also included this question as a limitation of the study.

 -  I recommend that the authors include biopac and visual stimuli in the procedures.

>>Response: Thank you for your suggestion. We have included this information in the Procedure subsection.

 -  I wonder if the visual stimuli, namely the sexual one, were previously validated.

>>Response: Thank you for your comment. We carried out a pilot study to ensure that visual stimuli with sexual content elicited sexual arousal. This information has been provided in the description of the Visual stimuli.

 -    Did the team consider allowing more time for the participant to rest before the experiment, or thought of having more time in the documentary film? After all the procedures of being in the lab and with the researcher, more than 3 minutes of a neutral film seems necessary. This may have a strong influence on the excitation response. I would like to know if this is considered one of the study’s limitations.

>>Response: Thank you for your comment. Before starting the experiment, the participant was at rest for 5 minutes to get used to the conditions of the experimental room and the placement of the devices. This information has been included in Procedure subsection.

-  The procedures must be clarified when the participants fill out the SIS/SES, GMREL, and GMSEX. Although this may be at the beginning of the online survey, it must be clarified.

>>Response: Thank you for the comment. This information is now provided in the Procedure subsection.

 -   I recommend the authors explain how the genital response was calculated.

>>Response: Thank you for your recommendation. This information now appears in the Procedure subsection.

 Discussion

-  The discussion becomes more difficult after line 227. The authors are sometimes redundant.

>>Response: Thank you for your indication. We have revised and modified this part to reduce redundances.

 -  The authors should be more coherent in the use of the terms sexual satisfaction and sexual satisfaction in their relationships throughout the paper.

>>Response: Thank you for your feedback. Terms have been revised to increase consistency throughout the document. When we use the term "sexual satisfaction" we are referring to the context of the couple.

-  Also, it is not clear why excitatory traits are not so related to sexual satisfaction. Clinical practice gives us other information if the literature fails to show this relationship. Also, the link of excitation with orgasm doesn’t explain the results and possible processes that underlie these results. I suggest that the authors go further in explaining these results, considering other processes related to inhibitory and excitation responses.

>>Response: Thank you for your comment.  We deepen in the Discussion section to explain the weaker relationship between the propensity for sexual excitation and sexual satisfaction. Also, we have added a clarification of the relationship between arousal, orgasm and sexual satisfaction.

-   I recommend adding other limitations, namely the ones related to the use of an experimental setting on the sexual response, the consideration of the trait, and the state decisions of the research team.

>>Response: Thank you for your recommendations. We have included limitations related with the experimental setting.

 -   The conclusion doesn’t explain why this study is essential, as the introduction does. I recommend the authors be more specific about how this study may contribute to further studies and, more specifically, to clinical interventions. Based on these results, what clinical strategies does the team suggest?

Response: Following your recommendation, we have expanded the conclusion for future research and clinical intervention.

 Again, we would like to sincerely thank the reviewer for the time spent in reviewing our manuscript and the constructive criticism to improve our work. Thank you very much!

Round 2

Reviewer 1 Report

Comments and Suggestions for Authors

The authors have done a great job in responding to my comments and suggestions on the original submission. I noticed some points that could further improve the paper.

The authors have added hypotheses, which I think is indeed important. They state the following ‘We expect that women who are more sexually satisfied in relationships compared to less satisfied women would (1) report lower propensity for sexual excitation [43] or not differ from each other [28,42], (2) report lower propensity for sexual inhibition [28,42,43], and (3) experience lower sexual arousal in the laboratory context (i.e., lower genital responsiveness and subjective sexual arousal to sexual stimuli) [44].’ Although most of the hypothesized effects can be traced back to the relevant references, the expectation that higher sexual satisfaction would associate with lower laboratory-induced sexual arousal is less obvious from the cited publication of Lawless and colleagues (2022) in my view. Instead, this association in the cited paper depends on the source of the sexual stimulation. Although the hypothesized negative association was found when women’s sexual arousal was induced by other than her own partner, sexual arousal induced by the woman’s partner was positively (and stronger, with r = .48) correlated with her sexual satisfaction. In the laboratory procedure in the present study it can be argued that the stimulation used is more representative of the ‘other than her own partner’, this should be made more explicit than is currently done.

Comments on the Quality of English Language

Page 1,  line 35: ‘According to the World Association for Sexual 34 Health (WAS), is considered an essential element …’ The term sexual satisfaction has probably been omitted here.

Page 2, lines 54-55: ‘… because of adequate sexual functioning …’ This not clear enough. Do you mean ‘ … and increases with adequate sexual functioning …’?

Page 7, lines 295-297: ‘It is noted the positive association between self-reported sexual arousal (i.e., RSA and RSG), but no with genital response, …’ The text here would benefit from language editing.

Page 8, line 324: ‘… that more general, …’ Here a noun following ‘general’ seems to be missing.

Page 8, line 325: ‘… inhibition DUE to …’

Page 8, lines 342-344: the text here requires micro-editing. It is now difficult to grasp the meaning of what is said.

Page 9, lines 368-369: ‘ … it is suggested to the importance of personal variables (i.e., propensity for sexual inhibition) in sexual satisfaction of women. ‘ Please edit to improve comprehensibility.

Author Response

The authors have added hypotheses, which I think is indeed important. They state the following ‘We expect that women who are more sexually satisfied in relationships compared to less satisfied women would (1) report lower propensity for sexual excitation [43] or not differ from each other [28,42], (2) report lower propensity for sexual inhibition [28,42,43], and (3) experience lower sexual arousal in the laboratory context (i.e., lower genital responsiveness and subjective sexual arousal to sexual stimuli) [44].’ Although most of the hypothesized effects can be traced back to the relevant references, the expectation that higher sexual satisfaction would associate with lower laboratory-induced sexual arousal is less obvious from the cited publication of Lawless and colleagues (2022) in my view. Instead, this association in the cited paper depends on the source of the sexual stimulation. Although the hypothesized negative association was found when women’s sexual arousal was induced by other than her own partner, sexual arousal induced by the woman’s partner was positively (and stronger, with r = .48) correlated with her sexual satisfaction. In the laboratory procedure in the present study it can be argued that the stimulation used is more representative of the ‘other than her own partner’, this should be made more explicit than is currently done.

>> Response: Thank you very much. Following your recommendations, we have clarified the hypothesis, and we have expanded the Procedure.

Comments on the Quality of English Language

Page 1,  line 35: ‘According to the World Association for Sexual 34 Health (WAS), is considered an essential element …’ The term sexual satisfaction has probably been omitted here.

Page 2, lines 54-55: ‘… because of adequate sexual functioning …’ This not clear enough. Do you mean ‘ … and increases with adequate sexual functioning …’?

Page 7, lines 295-297: ‘It is noted the positive association between self-reported sexual arousal (i.e., RSA and RSG), but no with genital response, …’ The text here would benefit from language editing.

Page 8, line 324: ‘… that more general, …’ Here a noun following ‘general’ seems to be missing.

Page 8, line 325: ‘… inhibition DUE to …’

Page 8, lines 342-344: the text here requires micro-editing. It is now difficult to grasp the meaning of what is said.

Page 9, lines 368-369: ‘ … it is suggested to the importance of personal variables (i.e., propensity for sexual inhibition) in sexual satisfaction of women. ‘ Please edit to improve comprehensibility.

>> Response: Thank you very much for your comments. All modifications related to the English quality have been carried out and marked in red in the manuscript.